# Passport to a Mighty Nation: Exploring Sociocultural Foundation of Chinese Public’s Attitude to COVID-19 Vaccine Certificates

**DOI:** 10.3390/ijerph181910439

**Published:** 2021-10-04

**Authors:** Mingyu Hu, Hepeng Jia, Yu Xie

**Affiliations:** School of Communication, Soochow University, Suzhou 215123, China; szhmy@suda.edu.cn (M.H.); xieyu2021@suda.edu.cn (Y.X.)

**Keywords:** COVID-19, vaccination passport, immunity certificates, nationalism, collectivism, CoV-SARS-2

## Abstract

Vaccination against COVID-19 is essential against the pandemic. There are broad discussions on adopting certificates for vaccination and the immunity obtained after infection. Based on a national sample of over 2000 participants administered in April 2021, the current study examines the Chinese public’s attitudes to the so-called COVID-19 vaccination passport and factors contributing to their viewpoints. Generally, the Chinese people had favorable opinions on the passport. Among possible contributing factors, income, personal benefit perception, the subjective norm of COVID-19 vaccination, and nationalism were significantly associated with the public’s positive attitude. At the same time, general vaccine knowledge and scientific literacy had an inconstant effect. Echoing recent studies, these findings reveal a collectivism-oriented attitude of the Chinese public towards the proposal to certify vaccination publicly. Theoretical and practical implications of the results were discussed.

## 1. Introduction

Amidst the rampant COVID-19 pandemic, scientists worldwide agree that massive vaccination against the epidemic is the only way to address the public health crisis [1,2]. Accompanying the global effort to improve the vaccination rate, governments and scientists propose the idea of COVID-19 immunity certificates—either for those fully vaccinated or for people having recovered from being infected by the virus—as an incentive to boost the public uptake of vaccination [3,4,5]. Indeed, European Commission began to adopt the digital COVID-19 certificate in July 2021 to facilitate travel across Europe [6].

Given the critical and controversial nature of the immunity certificate [7], many studies have examined the public attitude to the certification and contributing factors [8,9,10]. However, no research has explored the opinions of the Chinese public on this issue, despite the crucial role the country has played both in fighting pandemics and in the global supply of the COVID-19 vaccine. The lack of China-specific data also resulted in negligence of the alternative sociocultural factors that might help explain the country’s specific approaches to promoting vaccination and dealing with the pandemic. The current study is an effort to fill this gap.

This study focuses on investigating people’s attitudes to viewpoints about vaccine passports (VP, online or paper proof of vaccination) instead of infection-based immunity due to China’s early success in controlling the COVID-19 pandemic and the correspondingly only a tiny percentage of the population being infected. Although China has not officially adopted any mandatory vaccine passport program, its health code—a mobile phone-based COVID-19 contact tracing tool [11]—has included vaccination information. Media stories and social media posts also show in some localities, mandatory vaccination was required to take public transportation [12].

For VP, existing studies found a moderate to high public support across countries, with people in the UK and the United States having the most favorite attitude, particularly for international travel [10,13,14]. Although varying across contexts, men and older populations generally support the passport [9,15], so do more educated and wealthy people [8].

In their regular survey, Baum and colleagues identified that people who have been vaccinated supported VP more than those who have not [8]. The finding necessitates an investigation into the link between factors contributing to people’s vaccination intention and their attitude to VP. Indeed, given the observed connection between the attitude to VP and vaccination intention [16], studies often examine the link between factors contributing to vaccination intention—such as increased COVID-19 concern and perceived virus severity—and attitude to VP.

Among these factors, knowledge—both issue-specific vaccine knowledge and more general scientific knowledge—is of particular interest to us, as vaccine knowledge and education level have been reported as positive predictors for vaccination willingness [17,18,19,20,21]. We thus expected that those with higher vaccine knowledge and scientific literacy (measured with both general scientific knowledge and analytical thinking) would be more supportive of VP.

While scholars found the perception of personal risks and benefits from vaccination to drive people’s acceptance of VP [9,21], Tsai argued that social factors might shape one’s attitude towards such a passport [22]. In oriental countries such as Japan and China, concern for others [9] and family members [23] are powerful predictors of getting vaccinated. For the current study, we test the relationship between attitude to VP and respondents’ perception of personal and family benefits from vaccination against COVID-19 to examine the role of societal factors. To assess the link between the perception of social acceptance of COVID-19 vaccination and attitude to VP, we also employ a widely used construct—subjective norm—in health behavior studies [24] to examine whether the perceived attitude of essential others toward vaccination may influence respondents’ viewpoints about VP.

To further examine the sociocultural actors shaping the Chinese public’s attitude towards VP, we finally test respondents’ perception of how COVID-19 vaccination benefits China’s national interest and their nationalism scores. Jia and Luo found nationalism was strongly associated with the Chinese public’s prevention intention against COVID-19 [25]. If any VP is administered in China, it is essentially a government initiative, while nationalism centrally measures people’s embrace of their government [26]. Given the rising nationalism discourses in China since the debut of the pandemic [27,28,29], it is meaningful to expect that people’s nationalism level should be associated with their attitude to VP. Similarly, their perception of the national benefit of COVID-19 vaccination should also play such a positive role.

## 2. Materials and Methods 

### 2.1. Study Design and Measures 

This study is based on a nationwide online survey administered during April 2021 by the Shanghai-based survey firm *Diaoyanba*. The current sample’s location, age cohorts, and gender distribution matched the population demographics in the China Statistical Yearbook 2019. Our school administration in a large research university in East China approved the research plan due to the lack of an institutional review board for social sciences. In the questionnaire, we stressed anonymity and privacy protection and allowed participants to exit any time if they felt uncomfortable. 

At the time of our survey, there was no new local COVID-19 infection in China, and the Chinese government was implementing the COVID-19 vaccination program nationally. However, as of 1 April 2021, China had only reported 11.82 million doses of COVID-19 vaccination across the country [30], which were far less than the total population of China (1.412 billion in 2020). 

The online questionnaire employed in this study aimed to assess the following five points: (1) The Chinese public’s attitude towards COVID-19 VP; (2) their demographic information; (3) their vaccination, scientific literacy, and vaccine knowledge; (4) their perception of personal and family benefits from COVID-19 vaccination and their subjective norm of the vaccination; (5) their perception of national benefits from COVID-19 vaccination and their nationalism score. 

Based on previous studies, we measured the attitude towards VP by asking participants on a five-point scale whether they agreed with claims collected from various debates regarding VP. We assessed the knowledge about vaccines and scientific literacy with an established instrument [31,32,33]. Based on recent studies [25,27,34], we developed an updated instrument to measure nationalism, including a well-established measure [26] and similar questions judging citizens’ loyalty to their government amidst the COVID-19 pandemic. We pre-tested the questionnaire, and the final Cronbach’s alpha of the survey was 0.92. We reported the survey questions in the tables below. 

### 2.2. Statistical Analysis

The statistical analysis was performed using SPSS Version 27.0 (IBM Corp, Armonk, NY, USA). When describing the Chinese public’s attitude to VP, we applied the Descriptive Statistics-Frequency function of SPSS. We analyzed the relationship between participants’ attitude to VP and their vaccination status, vaccine knowledge level and scientific literacy, benefit perception, and nationalism score with the Hierarchical Linear Regression function of SPSS. A *p*-value < 0.05 was considered statistically significant.

## 3. Results

### 3.1. Demographic Characteristics

The demographic breakdown of the studied group is presented in Table 1. Most of those surveyed were middle-aged (40–49, 26.6% and 30–39, 25.9%), male (51.3%), completed junior college education (34.0%) and college degree education (33.8%), and earned CNY 3001–5000 (36.2%) or CNY 5001–10,000 (30.6%) per month (USD 1 = CNY 6.48).

### 3.2. Attitude to Vaccination Passport

We selected a five-point Likert-type scale to examine the level of the respondents’ attitude to VP (Table 2). The measure, consisting of the four questions reported in Table 2, was adapted from recent literature [9,20]. Cronbach’s alpha for the four questions was 0.83. The survey data show that the respondents generally had a very positive attitude towards VP. An average of 29.91% agreed or totally agreed to the statements related to VP, while only 8.28% disagreed or totally disagreed to them. 

### 3.3. Vaccination, Scientific Literacy, and Vaccine Knowledge

As this survey was administered in early April 2021, most of our sample (*n* = 1890) has not been vaccinated. We set to examine whether having the COVID-19 shot was associated with the attitude to VP.

This survey also investigated participants’ scientific literacy and vaccine knowledge. Based on well-established instruments used worldwide on scientific literacy [32], we selected 11 questions from the pool to measure respondents’ scientific literacy. We used six questions from Zingg and Siegrist to measure their vaccine knowledge [33].

Most questions consisted of three options: the statement is ‘wrong,’ ‘correct,’ or ‘I don’t know’ (Table 3). One point was assigned only for the correct answer. Respondents had relatively high scientific literacy and good vaccine knowledge. As reported in Figure 1, those with a score of 6 or above in scientific literacy level accounted for 59.9% (*n* = 1221), and respondents with a score of 3 or above in vaccine knowledge level accounted for 66.5% (*n* = 1355). 

### 3.4. Perceptions of Personal and Family Benefits, Subjective Norm 

We used five-point Likert-type scales to measure participants’ perception of personal and family benefits from vaccination (Table 4). Participants had a high agreement to the personal benefit from vaccination (47.5%) and a very high endorsement of COVID-19 vaccines’ protection role for family members (85.1%). 

The survey also measured the subjective norm of vaccination with an instrument consisting of two conventional questions to reflect people’s perception of how others surrounding them thought about immunization. The measure consisted of the two questions was reported in Table 5. The Cronbach’s alpha for the two questions was 0.82. Respondents generally had a strong perception of other people’s endorsement of the COVID-19 vaccination. 

### 3.5. Perception of National Benefits from Vaccination and Nationalism

Five-point Likert-type scales were selected to measure the level of the respondents’ perception of national benefits from vaccination (Table 6). Cronbach’s alpha for the two questions—one for general national benefits and one for national economic return resulting from the epidemic control—was 0.84. Generally, participants had a very high endorsement of the national benefits from vaccination. 

Finally, we used an updated instrument to examine nationalism [25,34] based on expanding a well-established measure [26]. The measure consisted of the six questions was reported in Table 7. Cronbach’s alpha for the six questions was 0.86.

### 3.6. The Association between Knowledge, Subjective Norms, Benefit Perception, Nationalism, and Public Attitude to VP

We used the hierarchical regression model to test the hierarchical correlation between demographic factors (gender, age, education, and income), knowledge factors (scientific literacy and vaccine knowledge), subjective norms on vaccination, benefit perception, nationalism, and public attitude to VP. Hierarchical regression enabled us to observe whether interested factors explain statistically significant variance in the dependent variable (DV) after accounting for all other relevant variables [35]. It is widely used in public health and sociology, psychology, and communication studies [36,37].

We reported the regression results in Table 8. After controlling for demographic variables, we found that vaccine knowledge in model 2 affected respondents’ attitude to VP. The more the Chinese public had vaccine knowledge, the more they are inclined to support VP. However, the significance disappeared when more variables entered models 3 and 4. 

Step 3 added the perceptions of personal and family benefits from vaccination and the subjective norm of the behavior to the regression model, explaining an additional 10.3% of the variation in the attitude to VP. The perceived personal benefit and subjective norm of vaccination were positively associated with respondents’ attitude to VP and remained significant in model 4. 

In step 4, the addition of nationalism significantly enhanced the model’s explanatory power by 9.1% of the variance. Nationalism most strongly affected respondents’ attitude to VP, while the national benefit perception was nonsignificant. 

## 4. Discussion and Conclusions

The findings partially confirm previous research on the public attitude to immunity certificates in the US, UK, and Japan [9,13]. For example, income and personal benefit perception were constantly correlated with attitude to VP, but as a whole, demographic variables that were significantly associated with attitude to VP in the UK, Germany, and Japan, such as gender, age, and education [9], were not significant predictors of such attitude in the Chinese context. The positive association between vaccination and support for VP did not appear in our survey either [8]. 

While in the Western context, attitudes towards the uptake of immunity passports are driven more by personal risks and benefits than societal factors [9], in China, the latter seemed to have played a more remarkable role. In our study, this was evidenced by the positive and robust association between subjective norms and nationalism and the attitude to VP. 

Nationalism was most strongly related to attitude. As nationalism measures the public’s pride in and loyalty to their nation, it is not strange to see that the more they scored high in nationalism, the more they tend to take actions directed at national achievements in controlling the pandemic. Although the Chinese government has not formally proposed VP, it has widely called on public vaccination against the virus. To the Chinese public, the link between VP and vaccination can be straightforward. Indeed, people’s level of nationalism was strongly associated with their intention to adopt protective behaviors against COVID-19 [24]. In our separate study, nationalism was also positively correlated with the Chinese public’s willingness to get vaccinated. Vaccination, mask-wearing, and the adoption of vaccination passports all seem to point to a national victory against the pandemic.

While the role of nationalism might be specific to the Chinese context, the Chinese situation is not a complete separation from other parts of the world. A recent study identified that collectivism predicts mask use during the COVID-19 pandemic in the United States and worldwide [38].

The subjective norm of vaccination’s robust association with the attitude to VP seems to reflect collectivistic orientation. Representing people’s subjective perception of other people’s endorsement of the desired behaviors, the subjective norm has been widely used to predict people’s intention for vaccination in various medical settings [39,40,41]. Understandably, if a person supports a policy, he or she certainly wants others nearby to take or endorse the action encouraged by the policy. In this sense, the subjective norm of vaccination also implies a tendency to follow the state’s directives and adopt collectivistic values.

By comparison, different from the literature [16], whether one was vaccinated was not significantly associated with the attitude to VP. This finding might imply that the individualistic value (whether I have a shot) may have given place to collectivistic thinking (whether others take a shot and support vaccination) in affecting people’s attitude to VP in the Chinese context. 

Unexpectedly, the perceptions of national and family benefits from vaccination were not significant predictors of the attitude to VP. The nonsignificance of national benefit perception reminded us that the nationalism-driven attitude to COVID-19 vaccination policy, including VP, might not be a utilitarian choice [42]. Instead, it is more likely to be an active adherence to the state rules. It is also possible because when this survey was administered, China had controlled the pandemic for nearly one year, the perception of vaccination benefits might not be firm at the national and/or family member level. 

The changes in the significance of vaccine knowledge and scientific literacy in their association with the attitude to VP are of interest for further discussion. The vaccine knowledge lost its significance in models 3 and 4 when subjective norm and nationalism were entered. The vaccine knowledge measure (e.g., vaccines lead to smallpox’s eradication and more resistance to diseases) likely involves recognizing the public health establishment shared by other variables in models 3 and 4, so that vaccine knowledge lost its significance.

In model 4, scientific literacy obtained significance to associate with the attitude to VP reversely. The more people were scientifically literate, the more they would oppose VP, though the association was fragile. The underlying mechanism is not completely clear. Perhaps scientifically literate persons are more likely to object to compulsory vaccination implied by the certification because they, being more familiar with the scientific process, were aware of the emergency feature of the available COVID-19 vaccines. This objection was significant only when nationalism counted, perhaps because those high in scientific literacy and nationalism score more strongly oppose mandatory vaccination. The number of such people must be pretty small, resulting in a fragile association.

As a whole, our studies reveal an assertive role of nationalism and collectivism-oriented values such as the subjective norm of vaccination in influencing the Chinese public’s attitude to vaccination passports. The finding is consistent with recent literature highlighting the importance of nationalism in the Chinese public’s preventive measures against COVID-19 [24]. To the Chinese public, both taking preventative measures and supporting vaccination policies seem not just for personal protection but also for helping the nation overcome the pandemic, making it a supreme power in the disease-stricken world. Although from a personal perspective, the Chinese public may not consciously connect their epidemic prevention behavior with the national benefits, the nationalism rooted in the Chinese people has indeed affected the adoption of their epidemic prevention behavior in the macro dimension.

While revealing the powerfulness of nationalism and other collectivism-oriented variables in affecting the Chinese public’s attitude to VP, this study also has a practical implication. In the Chinese context, stressing national demand and providing clear policy guides might be persuasive framing delivered in public health campaigns, just as what has been done historically during the 1950s’ Patriotic Hygiene Movement [43].

It is also necessary to discuss the consequences of adopting VP from the perspective of international comparison. A study comparing Israel (where VP was adopted) and UK respondents found that domestic vaccine passports may negatively affect people’s autonomy, motivation, and willingness to get vaccinated [44]. The finding was consistent with the suggestion that besides vaccine certificates, public health officials should also trigger people’s intrinsic motivation to address the pandemic [45]. 

Although our data cannot reveal whether Chinese people’s attitude to VP can influence their vaccination intention, it seems that VP adoption should not have a remarkable detrimental effect on people’s autonomous motivation to get vaccinated. Chinese people may consider both VP and vaccination are to meet the national demand to fight the pandemic. Correspondingly, public health campaigns may need to focus on the national demand frame to trigger people’s intrinsic motivation in the Chinese context. 

### Strengths and Limitations

Despite its finding on the China-specific contributors to people’s attitude to VP, this study is not without limitations. First, this study is mainly based on events in the Chinese context. Therefore, the main conclusion of our research, the decisive role of nationalism and subjective norm of vaccination in influencing people’s attitude to the COVID-19 vaccination passport, may hardly be generalized. Indeed, in more regular healthcare settings such as taking a flu shot, nationalism may not play a role as vital as in the pandemic setting because the strong nationalism might be activated by the remarkable achievement China has made during its fight against the pandemic. Given the uniqueness of the Chinese sociopolitical context and the COVID-19 setting, we would not recommend the attempt to generalize our findings. Instead, our study further confirms that the factors influencing public attitudes towards COVID-19 vaccines were embedded in different socio-ecological contexts [46]. 

Second, the survey’s cross-sectional feature and its relatively small sample size compared with China’s huge population restrict us from making any causal conclusion. Although based on the data available to us, we cannot wholly exclude possible confounding factors [47], the fact that several studies have set up links between the public’s collectivistic/nationalistic orientation and the intention/behavior against the COVID-19 pandemic safeguards our major finding on the association between Chinese people’s nationalism score and their attitude to VP [25,38]. The nature of a cross-sectional survey administered in a relatively short time determines that we cannot diagnose exactly where the confounding factors may exist. However, considering the previous studies and specific COVID-19 situation in China where there is a great pride of national triumph against the pandemic and few confirmed cases of infections, such confounding factors are very likely to link to the country’s sociocultural situation instead of demographic and pathological elements. Follow-up research with a more delicate design, broader context, and more diversified research methods might help address the limitations of the context and cross-sectional nature of the current survey-based research. 

## Figures and Tables

**Figure 1 ijerph-18-10439-f001:**
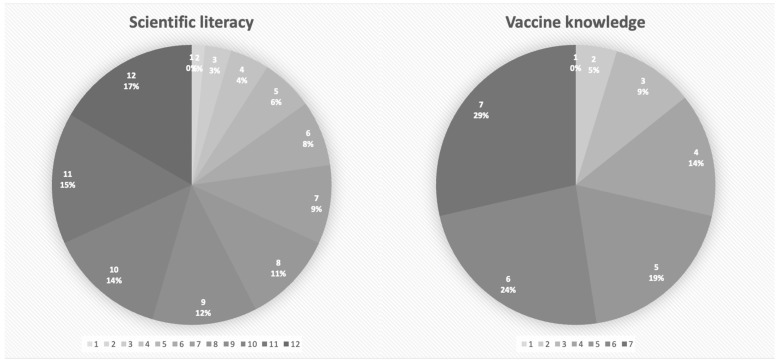
Level of scientific literacy and vaccine knowledge of participants (*n* = 2038).

**Table 1 ijerph-18-10439-t001:** Distribution of demographic characteristics of the sample (*n* = 2038).

Variable	% (*n*)
Age	
18–29	21.7 (443)
30–39	25.9 (527)
40–49	26.6 (542)
50–59	25.8 (526)
Gender	
Male	51.3 (1045)
Female	48.7 (993)
Education level	
Junior high school and below	12.4 (253)
Senior high school	17.5 (356)
Junior college education	34.0 (692)
College degree	33.8 (688)
Postgraduate and above	2.4 (49)
Monthly income	
3000 or less	25.1 (511)
3001–5000	36.2 (738)
5001–10,000	30.6 (623)
10,001–20,000	7.0 (142)
More than 20,000	1.2 (24)

**Table 2 ijerph-18-10439-t002:** Participants’ attitudes to VP (*n* = 2038).

Question: Please Make Your Judgment on the Following Statements on COVID-19 Vaccine Management:					
	Totally Disagree	Disagree	Neutral	Agree	Totally Agree
(1) The government should let vaccinated people have a VP so that they can freely move if there is (new round of COVID-19) epidemic;	17.6% (358)	10.0% (204)	26.1% (532)	18.1% (369)	28.2% (575)
(2) After being vaccinated, people should show VP when traveling across cities;	7.6% (154)	6.8% (138)	24.1% (491)	21.8% (445)	39.7% (810)
(3) Vaccination record should be treated as a health code to facilitate inspection;	7.6% (154)	6.8% (138)	24.1% (491)	21.8% (445)	39.7% (810)
(4) The government has the right to force people to show COVID-19 vaccination records publicly when necessary	5.0% (102)	4.8% (98)	20.2% (412)	20.2% (412)	49.8% (1014)

**Table 3 ijerph-18-10439-t003:** Participants’ scientific literacy and knowledge about vaccine (*n* = 2038).

General Scientific Questions			
**Question:** Two scientists want to know whether a drug for high blood pressure is effective. The first scientist distributed the drug to 1000 patients with hypertension and then observed how many patients had their blood pressure decreased; The second scientist divided the patients into two groups. The first group of 500 patients with hypertension took medicine, while the other 500 patients did not take medication. Then He observed how the blood pressure decreased in the two groups. Which of the first and second scientists is more effective in testing the effect of drugs?	**First One**	**Second One**	**I Don’t Know**
	14.7% (299)	70.0% (1427)	15.3% (312)
**Question:** The doctor told a couple that because they have the same morbid genes, if they give birth to a child, their chance of genetic disease is 1/4. Do you think the following statement is correct?	**Wrong**	**Correct**	**I don’t know**
If they have three children, none of them will get genetic diseases.	62.1% (1266)	8.1% (166)	29.7% (606)
If their first child has a genetic disease, the subsequent three children will not have a genetic disease.	62.2% (1268)	8.9% (181)	28.9% (589)
If the first three children are healthy, the fourth child must have a genetic disease.	60.2% (1226)	8.9% (181)	31.0% (631)
Their children may have genetic diseases.	15.0% (306)	57.0% (1162)	28.0% (570)
**Question: Do you think the following statement is correct?**			
Cov-SARS-2 can cause SARS and pneumonia, but it will not cause colds.	56.3% (1147)	9.7% (197)	34.1% (694)
Electrons are smaller than atoms.	23.3% (475)	31.6% (643)	45.1% (920)
The mother’s genes determine whether the child is a boy or a girl.	72.1% (1469)	7.5% (153)	20.4% (416)
Lasers are produced by converging sound waves.	22.7% (462)	18.1% (368)	59.3% (1208)
Antibiotics (such as penicillin, streptomycin, or cephalosporin) can kill both bacteria and viruses.	39.9% (814)	24.9% (507)	35.2% (717)
If you eat genetically modified fruit, human genes may change.	50.4% (1028)	15.4% (314)	34.2% (696)
**Vaccine-related questions**			
We don’t necessarily need a vaccine because diseases can always be cured.	78.0% (1590)	2.4% (49)	19.6% (399)
Smallpox will not be eradicated unless vaccines are widely used.	16.0% (327)	57.5% (1172)	26.4% (539)
If many vaccines are given too early, children’s immune systems will not develop normally.	45.5% (928)	16.3% (333)	38.1% (777)
Vaccination does not increase the incidence of allergies.	36.6% (745)	15.3% (312)	48.1% (981)
If children are not vaccinated so much, they will be more resistant to diseases.	62.6% (1276)	9.0% (184)	28.4% (578)
Autism, multiple sclerosis, and diabetes may be caused by vaccination.	54.1% (1102)	8.0% (163)	37.9% (773)

**Table 4 ijerph-18-10439-t004:** Participants’ perception of personal and family benefits from COVID-19 vaccination (*n* = 2038).

Question: To What Extent Do You Agree with the Following Statements?					
	Totally Disagree	Disagree	Neutral	Agree	Totally Agree
**Perception of Personal and Family Benefits**					
COVID-19 vaccine will keep me from getting sick caused by the epidemic.	1.1% (23)	6.1% (125)	45.2% (922)	34.7% (707)	12.8% (261)
COVID-19 vaccination can protect my family from the risk of the epidemic.	1.6% (32)	1.7% (35)	11.7% (238)	51.3% (1045)	33.8% (688)

**Table 5 ijerph-18-10439-t005:** Participants’ subjective norms (*n* = 2038).

Question: To What Extent Do You Agree with the Following Statements?					
	Totally Disagree	Disagree	Neutral	Agree	Totally Agree
**Subjective Norms for COVID-19 Vaccination**					
People around me think I should get a COVID-19 vaccine	8.9% (182)	4.8% (98)	20.6% (420)	18.8% (384)	46.8% (954)
People around me are willing to take the COVID-19 vaccine	4.5% (92)	4.0% (81)	19.6% (399)	20.5% (418)	51.4% (1048)

**Table 6 ijerph-18-10439-t006:** Participants’ perception of national benefits from vaccination (*n* = 2038).

Question: To What Extent Do You Agree with the Following Statements?					
	Totally Disagree	Disagree	Neutral	Agree	Totally Agree
**Participants’ Perception of National Benefits from Vaccination**					
Extensive vaccination of the COVID-19 vaccine can protect the country from the pandemic	1.5% (30)	3.0% (62)	21.8% (445)	48.9% (996)	24.8% (505)
Extensive vaccination of the COVID-19 vaccine can protect China’s economy from the pandemic	2.0% (41)	4.8% (97)	24.8% (505)	46.1% (940)	22.3% (455)

**Table 7 ijerph-18-10439-t007:** Participants’ nationalism score (*n* = 2038).

Question: To What Extent Do You Agree with the Following Statements?							
	Totally Disagree	Disagree	Slightly Disagree	Neutral	Slightly Agree	Agree	Totally Agree
**Participants’ Nationalism Score**							
I would rather become a citizen of China than of other countries	2.7% (56)	1.5% (30)	2.2% (44)	5.1% (103)	5.7% (116)	7.1% (144)	75.8% (1545)
My country is better than most other countries	2.1% (43)	1.2% (24)	1.8% (36)	5.1% (104)	6.1% (125)	7.9% (160)	75.9% (1546)
We should support our government even if it is wrong	19.2% (391)	7.0% (143)	13.9% (284)	22.0% (449)	12.9% (263)	6.4% (131)	18.5% (377)
China performed better than most other countries in controlling COVID-19	1.9% (39)	1.0% (20)	2.4% (49)	5.2% (105)	5.4% (111)	7.6% (154)	76.5% (1560)
China performed better than most other countries in controlling COVID-19	1.7% (34)	1.0% (20)	2.3% (46)	6.0% (123)	6.1% (124)	8.1% (165)	74.9% (1526)
China performed better than most other countries in controlling COVID-19	2.5% (51)	1.2% (25)	2.8% (57)	9.0% (184)	9.7% (197)	10.5% (214)	64.3% (1310)

**Table 8 ijerph-18-10439-t008:** Hierarchical regression model of factors associated with attitudes to vaccination passport (*n* = 2038).

Step	Variable	Public Attitude to VP			
		Model 1	Model 2	Model 3	Model 4
		*β*	*β*	*β*	*β*
1	Demographic factors				
	Gender (ref. female) Male	0.037	0.041	0.026	0.028
	Age	0.023	0.019	0.009	−0.004
	Education (ref. primary/lower secondary)				
	Upper secondary	0.011	−0.004	−0.012	−0.006
	Junior college	−0.008	−0.018	−0.025	−0.035
	Undergraduate degree	−0.054	−0.028	−0.037	−0.047
	Postgraduate degree	−0.033	−0.043	−0.047 *	−0.036
	Income	0.086 **	0.080 **	0.054 *	0.055 *
2	Vaccination and knowledge factors				
	Whether vaccination (ref. No)		0.015	0.000	0.013
	Scientific literacy		0.013	0.015	−0.045 *
	Vaccine knowledge		0.128 ***	0.033	0.018
3	Benefit perceptions and subjective norms				
	Perceptions of personal benefits			0.111 ***	0.079 **
	Perceptions of family benefits			0.023	−0.030
	Subjective norms of vaccination			0.279 ***	0.209 ***
4	National benefits and nationalism				
	Perceptions of national benefits from vaccination				0.036
	Nationalism				0.326 ***
	Model statistics				
	Adjusted R^2^	0.008	0.025	0.127	0.218
	ΔR^2^	0.011	0.018	0.103	0.091
	ΔF	3.337 **	12.559 ***	80.097 ***	119.172 ***
	Model F	3.337 **	6.143 ***	23.763 ***	38.889 ***

* *p* < 0.05. ** *p* < 0.01. *** *p* < 0.001.

## Data Availability

Materials and anonymous data are available from the authors by request.

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
