# Peer review of "Passport to a Mighty Nation: Exploring Sociocultural Foundation of Chinese Public’s Attitude to COVID-19 Vaccine Certificates"

_ijerph, 2021, doi:10.3390/ijerph181910439_

Round 1

Reviewer 1 Report

The paper is interesting and covers a current issue in an underexplored context. The introduction sets the scene well, methods are appropriate and the authors present appropriate limitations to position their overall claims. The only minor comment that I have is that in the discussion section, it would have been useful to expand on the comparisions between the findings in China and other studies conducted in the UK, etc. 

Author Response

Thank you for providing the instructive guides, which are crucial to improving the quality of the current manuscript. As you may find from our revision, we have now added two comparisons. First, based on studies we already reviewed, we extended discussions to compare the difference between our findings and those of these studies. The key difference seems to be an individualistic choice (in the UK, US, and other European nations) and collectivistic demand (in China), which is consistent with the overall theme of our study.

Second, we added a new reference examining the effect of vaccine passports in Israel and the UK, which found that domestic vaccine passports may negatively affect people's autonomy, motivation, and willingness to get a vaccination. By comparing the different socio-cultural situations revealed from our data, we suggested that adopting VP in China may not have a detrimental effect.

We wish our revisions and replies are acceptable. If you have new suggestions, please kindly inform us through the reviewing system. We are delighted to address them to improve our draft.

Reviewer 2 Report

In the study of Hu et al. entitled “Passport to a Mighty Nation: Exploring Sociocultural Foundation of Chinese Public’s Attitude to COVID-19 Vaccine Certificates” the authors conducted a study based on a national sample of over 2,000 participants administered in April 2021, to know the Chinese public's attitudes to COVID-19 vaccination passport and factors contributing to their viewpoints. Generally, the Chinese people had favorable opinions on the passport. Among possible contributing factors, income, personal benefit perception, the subjective norm of COVID-19 vaccination, and nationalism were significantly associated with the public's positive attitude. At the same time, general vaccine knowledge and scientific literacy had an inconstant effect. Echoing recent studies, these findings reveal a collectivism-oriented attitude of the Chinese public towards the proposal to certify vaccination publicly. The theme is important and can contribute to the discussion for the use and adoption of the COVID-19 vaccination passport around the world. For these reasons, the manuscript should be accepted after minor revisions.

Minor revisions:

1) Page 3, line 97: Change to 1.412 billion in 2020 (http://www.stats.gov.cn/english/PressRelease/202105/t20210510_1817185.html)

2) Page 4, lines 137 to 138: Since most people were not vaccinated, couldn't the expectation for the vaccine have influenced the response? That is, a person who has not yet been vaccinated would be more likely to answer something positive about the obligation to use the COVID-19 vaccination passport.

3) Page 6, Table 4: Question 1) COVID-19 vaccine will keep me from contracting the epidemic. Couldn't it be better worded? Vaccination does not prevent infection, but it does prevent the person from getting sick. So, a more appropriate question would be: Will the COVID-19 vaccine decrease the chance of me being hospitalized or dying?

4) Page 8, Discussion. Authors should include a subtopic of "Strengths and Limitations" similar to the study by Porat et al., 2021 (https://doi.org/10.3390/vaccines9080902)

Author Response

Thank you for providing the instructive guides, particularly those correction suggestions, which are crucial to improving the quality of the current manuscript. We have revised the text accordingly. You can find the revisions in red. Concretely, we are addressing your guides below:

1) Page 3, line 97: Change to 1.412 billion in 2020 (http://www.stats.gov.cn/english/PressRelease/202105/t20210510_1817185.html)

AUTHORS: Sorry for the typo. Thank you for getting this out for us. We have corrected the number.

2) Page 4, lines 137 to 138: Since most people were not vaccinated, couldn't the expectation for the vaccine have influenced the response? That is, a person who has not yet been vaccinated would be more likely to answer something positive about the obligation to use the COVID-19 vaccination passport.

AUTHORS: Thank you for pointing out this. Vaccination may likely influence response, as our literature suggests (ref. 8, Baum et al. 2021). We indeed examined this influence by setting vaccination as a variable, and the result is there was no significant association between vaccination and Chinese respondents' attitude to vaccine passport. Based on this fact and others, we suggested in our Discussion that the non-significance is very likely to be the evidence of the collectivistic orientation of the Chinese respondents. Please refer to lines 251-255 on Page 9. Thank you.

3) Page 6, Table 4: Question 1) COVID-19 vaccine will keep me from contracting the epidemic. Couldn't it be better worded? Vaccination does not prevent infection, but it does prevent the person from getting sick. So, a more appropriate question would be: Will the COVID-19 vaccine decrease the chance of me being hospitalized or dying?

AUTHORS: Thank you for pointing out this. Our original questionnaire is in Chinese. It seems to have indirectly included the type of meaning you suggested. We also revised the question mentioned in Table 4 in our manuscript as indicated to make the text more understandable.

4) Page 8, Discussion. Authors should include a subtopic of "Strengths and Limitations" similar to the study by Porat et al., 2021 (https://doi.org/10.3390/vaccines9080902).

AUTHORS: Thank you for this suggestion, which also helped us notice the exciting study by Porat et al. In the revision, we have added the "Strengths and Limitations" section and expanded our original discussions. Indeed, we also cited Porat et al. 2021 to expand our discussion on the consequences of VP.

Reviewer 3 Report

The article, even though it brings to light a current issue, which is the
permission to walk freely in different sectors and countries,
the article has little robustness in terms of sample and methodology.
Furthermore, there are several confounding factors due to the current situation
of COVID-19 around the world. China has a different socio-cultural situation
in relation to the USA and the UK,
which may have different implications for the PV.
It is important to better assess the confounding factors of the article
for publication

Author Response

Thank you for providing the crucial instructions. We have met several times and re-examined our data. Below we will reply to your instructions while reporting the manuscript revisions to address your concerns.

  1. About the background and relevant references:

AUTHORS: We have slightly added some new references to streamline our reasoning and expand our discussions, such as citing Porat et al., 2021 (https://doi.org/10.3390/vaccines9080902) to add the discussions on the consequences of adopting vaccine passport (VP) and the possible difference between China and Western countries where VP attitude data were available.

  1. The robustness in terms of sample and methodology.

AUTHORS: This is an essential and legitimate concern. We tried to address it very seriously. It is a national sample collected by quota sampling from a survey company's sampling pool. Respondents were  selected from the sampling pool. Our sample's regional (provincial) location, age cohorts, and gender distribution matched the population demographics in China Statistical Yearbook 2019. Since it's challenging to get a nationally representative and probability-based sample, we think the national sample collected by quota sampling was somehow a proper substitute. It may not have been able to reflect every Chinese person's viewpoints about VP representatively. Still, it certainly reflects a typical Chinese phenomenon: the collectivistic and nationalistic thinking somewhat dominated people's thinking regarding vaccine management (VP is part of the vaccine management regime in our questionnaire context).

For the methodology, it is correct that a public survey and regression analysis cannot exclude confounding factors, neither can it reach causality. We will address this below together with the confounding factor issue.

  1. Confounding factor issue.

AUTHORS: This is truly a critical issue, and we have to admit that we hadn't planned very carefully for this issue when designing this survey. After re-examining the data based on the suggestion of literature on confounding factors (e.g., Pourhoseingholi, M. A., Baghestani, A. R., & Vahedi, M. (2012). How to control confounding effects by statistical analysis. Gastroenterology and hepatology from bed to the bench, 5(2), 79-83.), we found it is impossible to identify them based on our data. However, we argue that the possible existence of founding factors should not deny the validity of our studies for the following justifications.

1) We have revised our texts throughout the manuscript to delete all causal expressions, such as dropping predict/predictors. The possible existence of confounding factors should not become an excuse to question the association relationship revealed by our data.  

2) We discussed the possible confounding factors even though we cannot diagnose precisely where these factors are based on our data. Because of our quota sampling and the few reported confirmed cases of infections, such confounding factors cannot be demographic and pathological. Instead, the possible confounding factors are likely to be those linked to the country's sociopolitical and socio-cultural context. For this, the association between Chinese people's nationalism score and their attitude to VP should still make sense despite the possible confounding effect. In addition, we can treat their socio-cultural background factors as consistent when facing Chinese respondents in the same social-cultural environment. Even though there are confounding factors caused by cultural background, they shouldn’t challenge the validity of this study’s main findings.

3) Finally, due to China’s unique situation in the COVID-19 particular circumstances, we will not extend our findings directly to other cultural contexts. We accordingly revised our description of the practical implications of our findings. 

Thank you once again for your insightful suggestions. In our follow-up research, we will carefully consider the confounding factor issue and avoid its impacts. If you have new suggestions, please kindly inform us through the reviewing system. We are delighted to address them to improve our draft.

Round 2

Reviewer 3 Report

The authors made new considerations regarding the article.
The methodology became more explicit and the discussion brought more
light and quality and clarity